# Sex differences in the association of vitamin D and metabolic risk factors with carotid intima-media thickness in obese adolescents

Indah K. Murni[1,2]*, Dian C. Sulistyoningrum[3], Danijela Gasevic[4], Rina Susilowati[5], Madarina Julia[1]

1 Department of Child Health, Dr. Sardjito Hospital/Faculty of Medicine, Public Health and Nursing, Universitas Gadjah Mada, Yogyakarta, Indonesia, 2 Center for Child Health—Pediatric Research Office, Faculty of Medicine, Public Health and Nursing, Universitas Gadjah Mada, Yogyakarta, Indonesia, 3 Department of Nutrition and Health, Faculty of Medicine, Public Health and Nursing, Universitas Gadjah Mada, Yogyakarta, Indonesia, 4 School of Public Health and Preventive Medicine, Monash University, Victoria, Australia, 5 Department of Histology and Cell Biology, Faculty of Medicine, Public Health and Nursing, Universitas Gadjah Mada, Yogyakarta, Indonesia

* indah.kartika.m@ugm.ac.id

**Data Availability Statement:** All relevant data are within the paper and its Supporting Information file.

## Abstract

### Background

It has been shown that vitamin D is associated with obesity and the development of atherosclerosis. Less is known about this association among adolescents with obesity.

### Objectives

To determine the association of vitamin D level and metabolic risk factors with carotid intima-media thickness (CIMT) among obese adolescents.

### Methods

We conducted a cross-sectional study among obese children aged 15 to 17 years in Yogyakarta, Indonesia. The association of vitamin D and other metabolic risk factors (triglyceride, low-density lipoprotein cholesterol (LDL-C) and high-density lipoprotein cholesterol (HDL-C), and insulin resistance using homeostasis model assessment of insulin resistance (HOMA-IR)) with CIMT was explored by multivariable linear regression models.

### Results

Out of 156 obese adolescents, 55.8% were boys. Compared to girls, boys had higher BMI z-score, waist circumference, and HDL-cholesterol. After adjustment for age, sex and second-hand smoke exposure, high HOMA-IR, total cholesterol, LDL-cholesterol and triglyceride levels were associated with higher odds of elevated CIMT. In analyses stratified by sex, a similar trend was observed in boys, while none of the risk factors were associated with CIMT in girls. We observed no association between vitamin D and CIMT.

**Funding:** we declare that the study was funded by the Indonesian Pediatric Society and Frisian Flag Indonesia (355/Legal/FFI/XII/2014). There was no additional external funding received for this study.

**Competing interests:** The authors have declared that no competing interests exist.

## Conclusions

Hyperinsulinemia, higher total cholesterol and LDL cholesterol were associated with greater odds of elevated CIMT among obese adolescent boys.

## Introduction

Atherosclerosis is a one of the important causes of cardiovascular and cerebrovascular diseases which lead to significant morbidity and mortality [1]. Subclinical atherosclerosis is an early indicator of atherosclerotic burden [2], and carotid intima-media thickness (CIMT) is regarded as a reliable marker of subclinical atherosclerosis since increased CIMT can reflect an increase in arterial wall thickness [3] including in children and adolescents [4]. Increased CIMT has been associated with obesity in children, familial hypercholesterolemia, type 1 diabetes, hypertension [5] and vitamin D deficiency [4].

Vitamin D is a steroid hormone with both endocrine and autocrine functions [6]. The endocrine function of vitamin D is mainly on the maintenance of calcium homeostasis and bone metabolism [7]. One of the autocrine functions of vitamin D is the modulation of inflammatory pathways which plays a role in cardiovascular diseases [8]. Vitamin D deficiency has been associated with the development of atherosclerotic and abnormality of arterial wall thickness mostly in adults [3, 9]. Vitamin D may play an important role in the endothelial or smooth muscle vascular cells and can also be involved in immune or inflammatory modulation [10, 11]. Therefore, vitamin D deficiency may contribute to an imbalance of vascular homeostasis, decreased arterial compliance, and the development of atherosclerosis [11]. Sufficient level of vitamin D in adults leads to a reduction in circulating inflammatory and endothelial function biomarkers; therefore, vitamin D might have a potential role as an anti-inflammatory therapy for the prevention and the treatment of cardiovascular disease [9]. Deficiency of vitamin D might be common in obese populations because of reduced sun exposure due to increased sedentary activities and sequestration of vitamin D in lipocytes [12]. There were few studies reporting the association between vitamin D deficiency with increased CIMT in children and adolescents with conflicting results [4, 13–15].

Studies evaluating the relationship of vitamin D deficiency and increased CIMT in obese adolescents are few and most studies were conducted in high-income countries among Caucasian populations [13–17]. Given limited data from low- and middle-income and tropical countries, studies are needed to better identify the association of vitamin D level and cardiovascular risk on vascular thickness among obese adolescents. This study aimed to determine the relationship of vitamin D status and other metabolic risk factors with subclinical atherosclerosis among obese adolescents in Indonesia (a low-to-middle income country with naturally abundant sun exposure). We were particularly interested in looking at CIMT as predicted by vitamin D status independent of other predictors of CIMT.

## Methods

### Study design and population

We conducted a cross-sectional study among adolescents in Yogyakarta, Indonesia. The recruitment process was done by screening for adolescents with obesity in seven public and three private high schools in Yogyakarta, a city in the Southern part of Java, Indonesia. The participants were recruited at their schools and we screened 4,268 students from 10 schools in

Yogyakarta using three body mass index (BMI) reference cut-off points: World Health Organization (WHO) [18], Centre for Disease Control and Prevention (CDC) [19], and International Obesity Task Force (IOTF) [20]. BMI was converted into BMI z-scores based on WHO Growth Reference 2007 using WHO AnthroPlus (https://www.who.int/growthref/tools/en/). The screening was done from January to February 2016. Obesity was considered when fulfilling all of the three obesity criteria [18–20]. Obese adolescents who agreed to participate in the next step of the study, i.e. assessment of their metabolic risk, were included in this study. The blood collection was taken at their high schools by a trained laboratory technician, while the assessment of carotid intima media thickness was performed at the Dr Sardjito Hospital, Yogyakarta. These data collections were done from March to October 2016.

Inclusion criteria were obese children aged 15 to less than 18 years who agreed to participate in this study. We excluded children with diabetes mellitus, renal disease, cardiovascular disease, any history of any systemic disease or history of current steroid use. Approval for the study was obtained from the Ethics Committee of the Faculty of Medicine, Universitas Gadjah Mada, Yogyakarta, Indonesia (KE/FK/333/EC/2016). Written informed consent was obtained from all parents/guardians included in the study.

## Data collection

All eligible children underwent history taking, physical examination, and blood collection. We collected demographic information as well as information on smoking exposures and family history of hypertension. We measured body weight, height, and waist circumference. The children's weight was measured using a portable weighing scale (CAMRY, EB9003) while they were in light clothing without shoes or slippers. The weight was recorded in kilograms (kg) to the nearest 0.1 kg. We measured height using a portable stadiometer or microtoise (GEA), and height was recorded in centimeters (cm) to the nearest 0.1 cm.

Waist circumference was measured using standardised procedures by placing a tape midway of the hipbone and the bottom of ribs and wrapping it around the child's waist. Abdominal obesity was defined as waist to height ratio $\geq$ 0.5 [21].

Blood pressure was reported as the average of three measurements collected after a 10-minute rest. Elevated blood pressure, including hypertension, was defined according to the Clinical Practice Guidelines for Screening and Management of High Blood Pressure in Children and Adolescents proposed by The American Academy of Pediatrics 2017. For adolescents aged $\geq$13 years, elevated blood pressure was systolic blood pressure of $\geq$120 mmHg, irrespective of diastolic blood pressure [22].

A total of 10 ml blood was collected to measure serum levels of triglyceride, low-density lipoprotein cholesterol (LDL-C), high-density lipoprotein cholesterol (HDL-C), fasting blood glucose, insulin, and glycated hemoglobin (HbA1c). Fasting plasma lipid profile was measured using enzymatic assays. Increased risk of diabetes and insulin resistance were assessed using HBA1c, fasting plasma glucose, fasting insulin and homeostasis model assessment of insulin resistance (HOMA-IR). Fasting plasma insulin was measured using immunoassay, while the fasting plasma glucose was measured using the hexokinase method. HOMA-IR was calculated from fasting plasma glucose and insulin using the specified formula: (fasting insulin (microU/L) x fasting glucose (nmol/L)/22.5) [23].

For impaired glucose metabolism, the criteria defined by the American Diabetes Association for increased risk of diabetes was fasting glucose $\geq$100 mg/dL and HbA1c $\geq$5.7% [24]. Insulin resistance was defined when HOMA-IR was >2.5 [24]. High insulin level was defined as insulin level above 15.7 microU/mL [25].

To define dyslipidemia, we used pediatric-specific cut-off values identified by the National Cholesterol Education Program (NCEP) Expert Panel on Cholesterol Levels in Children for: high triglyceride ≥130 mg/dL, high total cholesterol ≥200 mg/dL, LDL cholesterol ≥130 mg/dL, HDL cholesterol <40 mg/dL [26].

Level of vitamin D was measured using enzyme-linked immunoabsorbent assay or ELISA (DRG® 25-Hydroxyvitamin D Total ELISA, EIA 5396), which had been validated in pediatric population [27]. Vitamin D status was defined as deficiency of serum vitamin D level at < 20 ng/mL, insufficiency of serum vitamin D level 20–30 ng/mL, and sufficient for serum vitamin D level at >30 ng/mL [4, 8].

Carotid intima-media thickness (CIMT) was measured using common carotid artery B-mode ultrasound [2]. The carotid arteries were imaged using a standard echocardiography machine of Phillips HD 15 with vascular probe L12.3. CIMT was measured on the posterior (far) wall of the left carotid artery at end diastolic phase. At least three measurements were taken at approximately 10 mm proximal to the bifurcation to derive mean CIMT. A trained technician who measured the carotid arteries was blinded to the subject's clinical information. Abnormal CIMT was determined if the value was ≥ 95th percentile for age, sex, and height. Elevated CIMT was defined when CIMT measurement ≥ 0.047 mm in girls and ≥ 0.049 mm in boys [28].

## Statistical analysis

Data analyses were performed using STATA version 12.1, StataCorp LP, Texas. Continuous data are presented as mean and standard deviation (SD) or median and quartiles 1 (Q1) and Q3, for normally distributed and skewed data, respectively. Categorical variables are presented as counts and percentages. Normality of variables was checked using Kolmogorov-Smirnov tests.

Normally distributed data are compared using independent sample t-test, while skewed data are compared using independent-samples Mann-Whitney U test. Categorical data are presented as percentages and compared using chi-square tests. The strength of correlation between vitamin D and metabolic risk factors with carotid intima-media thickness between boy and girl obese adolescents were analyzed using Pearson correlation.

The dependent variable was CIMT as a marker of early vascular impairment. The independent variables were vitamin D and other metabolic risk factors including waist-to-hip ratio (marker of abdominal obesity), blood glucose, HbA1c (impaired glucose metabolism), fasting insulin, HOMA-IR (insulin resistance), LDL and HDL cholesterol, triglyceride (impaired lipid metabolism), blood pressure, smoking and family history of hypertension.

The association of vitamin D and other metabolic risk factors with CIMT was explored using logistic regression and presented as odds ratio (OR) with 95% confidence interval (CI). Each factor was tested in a separate regression model. Models that were significantly associated with elevated CIMT, were re-tested with adjustment for age, sex and secondary smoke exposures. All analyses were performed on a sample as a whole and also stratified by sex based on biological differences in body fat accumulation and potentially vitamin D levels between boys and girls.

## Results

We screened 4,268 students from seven public and three private high schools in Yogyakarta, Indonesia and identified 298 (7%) adolescents classified as obese based on all three obesity criteria of World Health Organization (WHO) [18], Centre for Disease Control and Prevention (CDC) [19], and International Obesity Task Force (IOTF) [20]. Of those, 229 (76.8%) obese

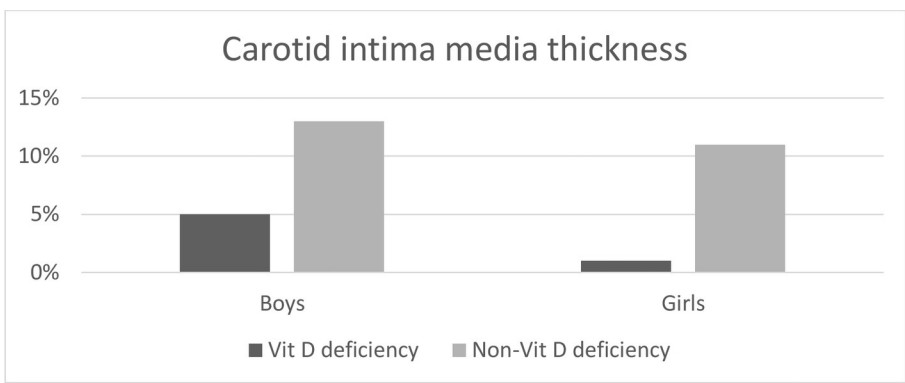

**Fig 1. Prevalence of increased carotid intima media thickness across vitamin D status among obese adolescents.**

adolescents agreed to participate in the next step of the study, i.e. assessment of their metabolic risk. The results of this assessment have been reported elsewhere [29].

This study included 156 (out of 229) obese adolescents who had data on CIMT. There were no differences in metabolic risk parameters between adolescents who had their CIMT measured versus those who had not (S1 Appendix). The exceptions were triglyceride levels that were on average higher among adolescents with data missing on CIMT. We considered that the sample of this study could be considered representative of a larger population.

Vitamin D deficiency (vitamin D at < 20 ng/mL) was observed in 30.1% of children, while 13.6% had vitamin D insufficiency (vitamin D level 20–30 ng/mL); and the rest of children (56.4%) had sufficient vitamin D level (vitamin D level at >30 ng/mL) (Fig 1).

Compared to girls, boys had significantly higher BMI z-score, waist circumference, and HDL cholesterol (Table 1). We found no correlation between vitamin D and CIMT, but LDL and total cholesterols were positively correlated with carotid intima media thickness among obese adolescent boys (Table 2). Boys with elevated CIMT showed increased values of LDL cholesterol (Table 3). Table 4 demonstrated that, overall, elevated HOMA-IR, total cholesterol, LDL-cholesterol and triglyceride levels were associated with greater odds of elevated CIMT. The associations persisted after adjustment for age, sex and second-hand smoke exposure. However, after the data were analyzed within each sex, similar results were only observed in boys. None of the risk factors were associated with CIMT in girls.

## Discussion

This study explored the relationship of vitamin D and other metabolic risk factors with CIMT in 156 obese adolescents in Indonesia. The results of the study indicate a positive association of hyperinsulinemia, hypercholesterolemia, and LDL hypercholesterolemia with CIMT among boys. No association was observed between vitamin D and CIMT among obese adolescents.

In our study, only about half of the Indonesian obese adolescents had normal vitamin D level, and vitamin D deficiency was observed in 30.1% of the adolescents. This is lower than the prevalence of vitamin D deficiency noted for adolescents aged 10–18 years from other countries (49% [30], 75% [31]); however, there are other studies that reported a lower prevalence of vitamin D deficiency than that observed in our study (4.5% [14], 20% [15]). We also observed the higher prevalence of vitamin D deficiency in boys compared to girls. In another study from Indonesia where researchers recruited 120 children between ages 7 and 12 years, there were 15% of adolescents with deficient vitamin D levels [32]. They observed that 75.8% of adolescents were with insufficient vitamin D levels, which was higher than that observed

**Table 1. Characteristics of between obese adolescent boys and girls from Yogyakarta, Indonesia.**

| Variables | All subjects | Boys | Girls | p value |
|---|---|---|---|---|
| | n = 156 (%) | n = 87 (%) | n = 69 (%) | |
| Age, years | 16.4 (0.7) | 16.3(0.7) | 16.5(0.6) | **0.030** |
| Weight, kg | 85.2 (78.0;95.7) | 91.7 (84.3;101.0) | 78.1 (73.0;84.6) | **<0.001** |
| Height, cm | 162.7 (9.2) | 168.6(7.0) | 155.3(5.5) | **<0.001** |
| BMI z-scores | 2.54 (2.30;2.90) | 2.62 (2.34;3.02) | 2.48 (2.26;2.74) | **0.004** |
| Waist circumference, cm | 93.3 (86.3;98.6) | 96.6 (92.3;104.5) | 86.1 (83.2;92.3) | **<0.001** |
| Waist-to-Height Ratio | 0.57 (0.54;0.60) | 0.58 (0.54;0.62) | 0.56 (0.53;0.59) | **0.007** |
| Systolic blood pressure, mmHg | 115.0 (106.0;124.0) | 119.0 (110.0;127.0) | 111.0 (102.0;120.0) | **<0.001** |
| Diastolic blood pressure, mmHg | 74.0 (69.0;81.0) | 73.0 (68.0;81.0) | 75.0 (70.0;82.0) | **0.143** |
| High waist-to-hip ratio, $\geq 0.5$ [a] | 151 (96.8) | 87 (100) | 64 (92.8) | **0.002*** |
| High waist-to-hip ratio, $\geq 0.6$ [a] | 39 (25.0) | 28 (32.2) | 11(15.9) | **0.020** |
| Vitamin D $\leq$30 ng/mL [b] | 68 (43.6) | 43 (49.4) | 25 (36.2) | 0.099 |
| Vitamin D $\leq$20 ng/mL [b] | 47 (30.1) | 31(35.8) | 16 (23.2) | 0.092 |
| Elevated fasting insulin, $\geq$ 15.7 mg/dL | 134 (86.5) | 78 (89.7) | 56 (82.4) | 0.187 |
| Elevated fasting blood glucose, $\geq$ 100 mg/dL | 6 (3.8) | 4 (4.6) | 2 (2.9) | 0.584* |
| HOMA-IR, >2.5 | 146 (96.2) | 83 (95.4) | 63 (92.6) | 0.467* |
| High HbA1C, $\geq$ 5.7% | 6 (3.8) | 3 (3.4) | 3 (4.3) | 0.772* |
| High total cholesterol, $\geq$ 200 mg/dL | 32 (20.5) | 14 (16.1) | 18 (26.1) | 0.125 |
| High LDL cholesterol, $\geq$ 130 mg/dL | 50 (32.1) | 28 (32.2) | 22 (31.9) | 0.968 |
| Low HDL cholesterol, < 40 mg/dL | 41 (26.3) | 29 (33.3) | 12 (17.4) | **0.025** |
| High triglyceride, $\geq$ 130 mg/dL | 58 (37.2) | 38 (43.7) | 20 (29.0) | 0.062 |
| Active smoking | 10 (6.4) | 10 (11.5) | 0 (0.0) | **0.002*** |
| Second hand smoke exposure | 105 (67.3) | 62 (71.3) | 43 (62.3) | 0.237 |
| Family history of hypertension | 47 (30.1) | 28 (32.2) | 19 (27.5) | 0.530 |
| Elevated CIMT ($\geq$0.47 mm in girls, $\geq$ 0.49 mm in boys) | 23 (14.7) | 15 (17.2) | 8 (11.6) | 0.323 |

Normally distributed and skewed continuous data are presented as mean (standard deviation or SD) and median (Quartile 1 or Q1;Q3), respectively. Normally distributed data are compared using independent sample t-test, while skewed data are compared using independent-samples Mann-Whitney U test. Categorical data are presented as n (%) and compared using chi-square tests unless otherwise indicated. P values report the significance comparing boys and girls.

* Fischer Exact Test.

[a] Different cut off was used: high waist-to-hip ratio $\geq 0.5$ or $\geq 0.6$.

[b] Different cut off was used: Vitamin D level $\leq$30 ng/mL or $\leq$20 ng/mL.

BMI = body mass index; HOMA-IR = homeostatic model assessment of insulin resistance; HbA1C = glycated hemoglobin, LDL = low density lipoprotein; HDL = high density lipoprotein; CIMT = carotid intima media thickness.

among our study participants (13.6%). They also observed the higher prevalence of vitamin D insufficiency in girls compared to boys, which is in contrast to the findings of our study [32]. This discrepancy in findings may be due to differences in study sample, whereby most of recruited subjects were girls (62.5%) in the other Indonesian study, while in our study the proportion of girls was 44.2%. We also only recruited obese adolescents, and mean BMI z-scores in boys (2.7) were higher compared to girls (2.5). Vitamin D has been considered to be lower in people with obesity because it is sequestered in their lipocytes [12].

We observed no association between vitamin D and CIMT in obese adolescents. This is consistent with the results of previously published studies in adolescents from Italy [14], and the United States [15, 16]. However, our results are in contrast to the studies from Turkey [4] and the United States [13], which noted a positive association between vitamin D deficiency and CIMT. We extended the results of previous studies by reporting a positive association of

**Table 2. Correlations between of vitamin D levels and other metabolic risk factors to predict carotid intima-media thickness in obese adolescent boys and girls.**

|  | Boys (n = 87) | | Girls (n = 69) | |
|---|---|---|---|---|
|  | ρ | p | ρ | p |
| Age, years | -0.027 | 0.801 | 0.044 | 0.717 |
| Waist-to-hip ratio | 0.119 | 0.273 | -0.144 | 0.237 |
| Systolic blood pressure, mmHg | 0.078 | 0.472 | 0.094 | 0.444 |
| Diastolic blood pressure, mmHg | -0.032 | 0.767 | 0.104 | 0.396 |
| Vitamin D, ng/mL | 0.077 | 0.481 | 0.163 | 0.182 |
| Fasting insulin, mg/dL | 0.104 | 0.339 | -0.127 | 0.303 |
| Fasting blood glucose, mg/dL | -0.139 | 0.198 | 0.082 | 0.501 |
| HOMA-IR | 0.088 | 0.420 | -0.125 | 0.309 |
| HbA1C, g/dL | -0.015 | 0.889 | 0.114 | 0.349 |
| Total cholesterol, mg/dL | 0.316 | 0.003 | -0.029 | 0.814 |
| LDL cholesterol, mg/dL | 0.337 | 0.001 | -0.012 | 0.920 |
| HDL cholesterol, mg/dL | -0.085 | 0.433 | -0.132 | 0.279 |
| Triglyceride, mg/dL | 0.152 | 0.160 | 0.105 | 0.392 |

LDL = low density lipoprotein; HDL = high density lipoprotein; HOMA-IR = homeostatic model assessment of insulin resistance; HbA1C = glycated hemoglobin.

LDL hypercholesterolemia, total hypercholesterolemia, hypertriglyceridemia and hyperinsulinemia with CIMT among obese adolescent boys. It is important to note that atherosclerosis can start in childhood showing lipid accumulation in the arterial intima [33]. Initially, children with atherosclerosis have at least some degrees of aortic fatty streaks [34, 35] and atherosclerotic plaques can be found in the coronary arteries during adolescence [35].

We observed no association between vitamin D and subclinical atherosclerosis in obese adolescents. It is possible that the lack of observing the association is due to vitamin D only serving as a marker for sunlight exposure [17, 36]. It is also likely that vascular changes may occur only after a significant period of vitamin D deficiency exposure, and therefore, the vascular effects may manifest later in children with chronic vitamin D deficiency [15]. Since we only measured vitamin D serum once, this may not reflect a long-term period of vitamin D deficiency. Therefore, our study population may have lacked time to develop atherosclerosis from vitamin D deficiency. In addition, a simultaneous measurement of inflammatory markers, parathyroid hormone, and estrogen with vitamin D level may strengthen the studies on the relationship of vitamin D and subclinical atherosclerosis [11, 37].

One study reported associations of vitamin D deficiency with adverse cardiovascular risk factors in children and adolescents since receptors of vitamin D can be found in vascular smooth muscles, endothelium, and cardiomyocytes [35]. Vitamin D deficiency contributes to the development of cardiovascular disease by promoting vascular stiffness and calcification, which lead to atherosclerosis [38]. Vitamin D deficiency is also associated with hypertension because of the renin-angiotensin-aldosterone system activation and endothelial system dysfunction [11, 39]. These can lead to the development of plaque as a degenerative vascular process that might result in myocardial infarction or stroke; and these processes start at a younger age [40].

The role of vitamin D deficiency in the development of atherosclerosis is also mediated by systemic and vascular inflammation [8, 11]. These include increased levels of inflammatory cytokines, such as C- reactive protein (CRP), tumor necrosis factor-α (TNF-α), and interleukin-6, and low levels of interleukin-10 [11]. The role of vitamin D on vascular smooth muscle

**Table 3. Differences of vitamin D levels and other metabolic risk factors in obese adolescent boys and girls with elevated carotid intima-media thickness and not elevated carotid intima-media thickness.**

| Variables | Boys (n = 87) | | | Girls (n = 69) | | |
|---|---|---|---|---|---|---|
| | Elevated CIMT n = 15 | Not Elevated n = 72 | p value | Elevated CIMT n = 8 | Not Elevated n = 61 | p value |
| Age, years | 16.15 (0.81) | 16.32 (0.67) | 0.394 | 16.56 (0.47) | 16.52 (0.61) | 0.849 |
| Systolic blood pressure, mmHg* | 116.0 (107.0;124.0) | 119.5 (108.5;126.0) | 0.749 | 115.5 (107.8;119.0) | 110.0 (102.5;119.0) | 0.343 |
| Diastolic blood pressure, mmHg* | 71.0 (69.0;80.0) | 72.0 (67.3;80.0) | 0.951 | 79.5 (72.0;84.5) | 74.0 (70.0;79.0) | 0.212 |
| BMI, kg/m$^2$ | 34.08 (4.51) | 32.67 (3.70) | 0.199 | 33.53 (3.86) | 32.64 (2.91) | 0.435 |
| BMI z-scores | 2.86 (0.49) | 2.67 (0.43) | 0.136 | 2.63 (0.46) | 2.53 (0.35) | 0.474 |
| Waist circumference, cm | 101.7 (10.6) | 97.9 (9.4) | 0.163 | 87.0 (7.4) | 86.7 (7.1) | 0.908 |
| Waist-to-Height Ratio* | 0.59 (0.55;0.64) | 0.57 (0.54;0.61) | 0.096 | 0.55 (0.52;0.60) | 0.55 (0.52;0.58) | 0.779 |
| Vitamin D, ng/mL* | 21.9 (16.0;51.3) | 30.9 (17.3;49.1) | 0.835 | 45.4 (32.4;57.0) | 36.9 (21.1;53.6) | 0.358 |
| Insulin, µIU/mL* | 35.6 (13.9;64.2) | 35.9 (24.0;55.2) | 0.991 | 24.7 (16.0;48.2) | 30.1 (19.5; 44.4) | 0.775 |
| Fasting Plasma Glucose, mg/dL | 84.80 (6.93) | 86.85 (6.63) | 0.283 | 89.38 (11.90) | 86.38 (15.74) | 0.606 |
| HOMA-IR* | 7.0 (2.8–11.4) | 6.6 (4.4;10.1) | 0.937 | 4.9 (3.0; 9.4) | 5.5 (3.9;7.9) | 0.805 |
| HbA1C, % | 5.26 (0.26) | 5.20 (0.27) | 0.448 | 5.25 (0.39) | 5.16 (0.98) | 0.801 |
| Cholesterol, mg/dL | 187.9 (24.9) | 172.6 (30.3) | 0.071 | 191.6 (47.0) | 174.8 (31.9) | 0.190 |
| LDL cholesterol, mg/dL | 131.9 (21.3) | 116.7 (27.9) | 0.050 | 133.0 (42.0) | 116.5 (29.5) | 0.160 |
| HDL cholesterol, mg/dL | 43.13 (7.43) | 42.76 (8.12) | 0.871 | 41.75 (11.94) | 48.43 (8.86) | 0.059 |
| Triglyceride, mg/dL * | 141.0 (95.0;173.0) | 118.5 (94.0;168.0) | 0.609 | 133.0 (83.5; 203.7) | 95.0 (73.5;133.5) | 0.097 |

Normally distributed and skewed continuous data are presented as mean (standard deviation or SD) and median (Quartile 1 or Q1;Q3), respectively. Normally distributed continuous data are compared using independent sample t-test, while skewed data are compared using independent-samples Mann-Whitney U test. Categorical data are presented as n (%) and compared using chi-square tests unless otherwise indicated. BMI = body mass index; HOMA-IR = homeostatic model assessment of insulin resistance; HbA1C = glycated hemoglobin, LDL = low density lipoprotein; HDL = high density lipoprotein; CIMT = carotid intima media thickness.

cells is also modulated by parathyroid hormones and estrogen [37]. Vitamin D and estrogen have already proven to help prevent Metabolic Syndrome and cardiovascular diseases in post-menopausal women [37].

We demonstrated the association of LDL hypercholesterolemia and hyperinsulinemia with CIMT in obese adolescent boys. This message could inform policy makers to formulate an effective prevention strategy for the development of cardiovascular diseases in obese adolescents. Vitamin D deficiency, obesity and its comorbidities and particularly high levels of insulin, HOMA-IR, LDL cholesterol, and total cholesterol should be prevented among adolescents, and boys especially, to avoid the development of cardiovascular disease.

The strength of our study is that it is among the first to evaluate the association of vitamin D and metabolic disease risk factors with vascular thickness in obese adolescents living in a country where sun exposure is abundant. However, the study is limited by its cross-sectional design and small sample size, and as a result, we may have not had enough power to detect the association between vitamin D and CIMT. In addition, since this study was only performed in the city of Yogyakarta, the results could not be generalized to other obese adolescents in Indonesia, or other low- and middle-income country settings with abundant sun exposure. More robust studies with larger sample sizes and using a longitudinal design are needed to further explore the association between vitamin D and subclinical atherosclerosis among obese adolescents living in low- and middle-income countries with abundant sun exposure.

**Table 4. The association between vitamin D levels and other metabolic risk factors to predict carotid intima-media thickness among obese adolescent boys and girls.**

| | All subjects (n = 156) | | Boys (n = 87) | | Girls (n = 69) | |
|---|---|---|---|---|---|---|
| | OR (95%CI) | *P* | OR (95%CI) | *p* | OR (95%CI) | *p* |
| Age, years | 0.76 (0.39–1.50) | 0.430 | 0.69 (0.30–1.60) | 0.390 | 1.13 (.32–3.99) | 0.846 |
| High waist-to-hip ratio, ≥0.5 [a] | na | Na | na | na | na | na |
| High waist-to-hip ratio, ≥0.6 [a] | 2.21 (0.87–5.60) | 0.096 | 2.13 (0.68–6.61) | 0.193 | 1.93 (0.34–11.08) | 0.463 |
| Elevated blood pressure | 1.03 (0.42–2.55) | 0.948 | 1.42 (0.46–4.40) | 0.545 | 0.70 (0.15–3.23) | 0.645 |
| Vitamin D ≤30 ng/mL | 1.01 (0.41–2.46) | 0.991 | 0.83 (0.27–2.52) | 0.740 | 1.82 (0.34–9.76) | 0.487 |
| Vitamin D ≤20 ng/mL | 1.66 (0.58–4.78) | 0.346 | 1.65 (0.48–5.70) | 0.429 | 2.28 (0.26–20.09) | 0.457 |
| Elevated fasting insulin, ≥ 15.7 mg/dL | 2.75 (0.94–8.06) | 0.065 | 4.87 (1.13–21.01) | **0.034**[d] | 1.67 (0.29–9.48) | 0.565 |
| Elevated fasting blood glucose, ≥ 100 mg/dL | 0.86 (0.10–7.71) | 0.859 | na | na | 0.12 (0.007–2.08) | 0.144 |
| HOMA-IR, >2.5 | 5.35 (1.32–21.69) | **0.019**[c] | 17.75 (1.70–185.1) | **0.016**[d] | 2.00 (0.20–20.51) | 0.559 |
| High HbA1C, ≥ 5.7% | 3.07 (0.53–17.83) | 0.211 | 2.50 (0.21–29.50) | 0.467 | 4.21 (0.34–52.65) | 0.264 |
| High total cholesterol, ≥ 200 mg/dL | 3.88 (1.51–9.96) | **0.005**[c] | 5.33 (1.50–18.94) | **0.010**[d] | 3.36 (0.74–15.18) | 0.116 |
| High LDL cholesterol, ≥ 130 mg/dL | 3.37 (1.36–8.36) | **0.009**[c] | 4.48 (1.31–13.33) | **0.015**[d] | 2.39 (0.54–10.61) | 0.252 |
| Low HDL cholesterol, < 40 mg/dL | 1.62 (0.63–4.15) | 0.319 | 1.00 (0.72–6.92) | 1.00 | 3.47 (0.70–17.11) | 0.127 |
| High triglyceride, ≥ 130 mg/dL | 2.54 (1.03–6.25) | **0.042**[c] | 2.22 (0.72–6.92) | 0.168 | 2.81 (0.63–12.59) | 0.176 |
| Active smoking | na | Na | na | na | na | na |
| Second hand smoke exposure | 1.45 (0.53–3.93) | 0.466 | 1.76 (0.45–6.86) | 0.416 | 1.01 (0.22–4.62) | 0.991 |
| Family history of hypertension | 1.61 (0.64–4.03) | 0.311 | 1.52 (0.48–4.78) | 0.478 | 1.69 (0.36–7.88) | 0.506 |

Abnormal conditions were given code = 1, normal = 0; LDL = low density lipoprotein; HDL = high density lipoprotein; HOMA-IR = homeostatic model assessment of insulin resistance; HbA1C = glycated hemoglobin; na = not applicable.

[a] Different cut off was used: high waist-to-hip ratio ≥ 0.5 or ≥ 0.6.

[b] Different cut off was used: Vitamin D level ≤30 ng/mL or ≤20 ng/mL, *≥0.47 mm in girls, ≥ 0.49 mm in boys.

[c] The association remained significant after controlling for age, gender and second hand smoke exposure.

[d] The association remained significant after controlling for age and second hand smoke exposure.

The dependent variable was CIMT as a marker of early vascular impairment.

The independent variables were vitamin D and other metabolic risk factors including waist-to-hip ratio (marker of abdominal obesity), blood glucose, HbA1c (impaired glucose metabolism), fasting insulin, HOMA-IR (insulin resistance), LDL and HDL cholesterol, triglyceride (impaired lipid metabolism), blood pressure, smoking and family history of hypertension.

## Conclusions

This study explored the relationship of vitamin D and metabolic disease risk factors with CIMT in obese adolescents in Indonesia. The results indicate a positive association of insulin resistance and dyslipidemia with CIMT among obese adolescent boys, while no association was observed between vitamin D and subclinical atherosclerosis in obese adolescents.

## Supporting information

**S1 Appendix. Characteristics of obese adolescents who had and had not CIMT measured.** (DOCX)

**S1 Fig. Prevalence of increased carotid intima media thickness across vitamin D status among obese adolescents.** (TIFF)

**S1 Data. Dataset of vitamin D and CIMT.** (XLSX)

## Acknowledgments

We would like to thank the Indonesian Pediatric Association Committee and Frisian Flag Indonesia (subsidiary of Friesland Campina) for funding this project. We also gratefully acknowledge Erik C Hookom for providing the editorial assistance.

## Author Contributions

**Conceptualization:** Indah K. Murni, Dian C. Sulistyoningrum, Danijela Gasevic, Rina Susilowati, Madarina Julia.

**Data curation:** Indah K. Murni, Dian C. Sulistyoningrum, Madarina Julia.

**Formal analysis:** Indah K. Murni, Danijela Gasevic, Madarina Julia.

**Funding acquisition:** Indah K. Murni.

**Investigation:** Indah K. Murni, Dian C. Sulistyoningrum, Rina Susilowati.

**Methodology:** Indah K. Murni, Dian C. Sulistyoningrum, Danijela Gasevic, Rina Susilowati, Madarina Julia.

**Project administration:** Indah K. Murni, Dian C. Sulistyoningrum.

**Resources:** Indah K. Murni, Dian C. Sulistyoningrum.

**Supervision:** Indah K. Murni, Dian C. Sulistyoningrum, Madarina Julia.

**Validation:** Indah K. Murni, Danijela Gasevic, Madarina Julia.

**Writing – original draft:** Indah K. Murni.

**Writing – review & editing:** Indah K. Murni, Dian C. Sulistyoningrum, Danijela Gasevic, Rina Susilowati, Madarina Julia.

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
