## [Decision Letter · Decision Letter 0]

30 Jul 2020

PONE-D-20-19921

SEX DIFFERENCES IN THE ASSOCIATION OF VITAMIN D AND METABOLIC RISK FACTORS WITH CAROTID INTIMA-MEDIA THICKNESS IN OBESE ADOLESCENTS

PLOS ONE

Dear Dr. Murni,

Thank you for submitting your manuscript to PLOS ONE. After careful consideration, we feel that it has merit but does not fully meet PLOS ONE’s publication criteria as it currently stands. Therefore, we invite you to submit a revised version of the manuscript that addresses the points raised during the review process.

While the study had a lot of merit, there were a number of issues that need to be addressed.  First their were significant grammatically errors throughout the text. Second, the data presented in the tables is unclear (what is being presented, skewed data is not indicated, and what was being compared). Third, why was Vitamin D not examined as a continuous variable? Also, from the raw data it appears that blood pressure was measured, why was this not included in the analysis? Lastly, as you did not find a association between vitD and CIMT, why was the analysis not extended to look at vitamin D to other metabolic risk factors measured.

We look forward to receiving your revised manuscript.

Kind regards,

Jonathan M Peterson, Ph.D.

Academic Editor

PLOS ONE

Journal Requirements:

3. In your Methods section, please provide additional information about the participant recruitment method and the demographic details of your participants. Please ensure you have provided sufficient details to replicate the analyses such as: a) the recruitment date range (month and year), b) a description of any inclusion/exclusion criteria that were applied to participant recruitment, c) a table of relevant demographic details, d) a statement as to whether your sample can be considered representative of a larger population, e) a description of how participants were recruited, and f) descriptions of where participants were recruited and where the research took place.

 "The study was partly funded by the Indonesian Pediatric Society and Frisian Flag Indonesia (355/Legal/FFI/XII/2014)".

i) Please provide an amended statement that declares *all* the funding or sources of support (whether external or internal to your organization) received during this study, as detailed online in our guide for authors at http://journals.plos.org/plosone/s/submit-now.  Please also include the statement “There was no additional external funding received for this study.” in your updated Funding Statement.

ii) Please include your amended Funding Statement within your cover letter. We will change the online submission form on your behalf.

Additional Editor Comments (if provided):

While the study had a lot of merit, there were a number of issues that need to be addressed. First their were significant grammatically errors throughout the text. Second, the data presented in the tables is unclear (what is being presented, skewed data is not indicated, and what was being compared). Third, why was Vitamin D not examined as a continuous variable? Also, from the raw data it appears that blood pressure was measured, why was this not included in the analysis? Lastly, as you did not find a association between vitD and CIMT, why was the analysis not extended to look at vitamin D to other metabolic risk factors measured.

Reviewers' comments:

Reviewer's Responses to Questions

**Comments to the Author**

1. Is the manuscript technically sound, and do the data support the conclusions?

Reviewer #1: Yes

2. Has the statistical analysis been performed appropriately and rigorously? 

Reviewer #1: No

3. Have the authors made all data underlying the findings in their manuscript fully available?

Reviewer #1: Yes

4. Is the manuscript presented in an intelligible fashion and written in standard English?

Reviewer #1: Yes

5. Review Comments to the Author

Reviewer #1: Paper has lot of grammatical errors and flow is very limited

Need to assess high sensitivity CRP/?cardiac risk/LVMI and 24 hr ABPM or BP and would have correlated these parameters with those variables

Vitamin d level should be correlated with these parameters as we do not know what level is optimal one

urine albumin to creatinine ratio should reassesses

I will assess height and growth parameters 'also

need to develop or assess cardiovascular risk score like in adult to assess risk prediction

we should look into family history

It is association not causation

6. PLOS authors have the option to publish the peer review history of their article (what does this mean?). If published, this will include your full peer review and any attached files.

Reviewer #1: No

---

## [Author Response · Author response to Decision Letter 0]

15 Sep 2020

Thank you for the constructive questions and feedbacks. Herewith our response to Reviewers.

Journal Requirements:

Response to the Reviewer:

Thank you, noted.

Response to the Reviewer:

The manuscript had been checked for the grammars and spellings by: 

Erik Christopher Hookom, BA, Med, TEFL

Office of Research and Publication (ORP)

Faculty of Medicine, Public Health, and Nursing Universitas Gadjah Mada

Administration Building 2nd Floor

Phone: 0274 560300 ext 205

Email: orp.fm@ugm.ac.id; echookom@gmail.com

A copy of manuscript showing track changes and a clean copy of the edited manuscript have been uploaded as a supporting information file and the new manuscript file, respectively. 

3. In your Methods section, please provide additional information about the participant recruitment method and the demographic details of your participants. Please ensure you have provided sufficient details to replicate the analyses such as: a) the recruitment date range (month and year), b) a description of any inclusion/exclusion criteria that were applied to participant recruitment, c) a table of relevant demographic details, d) a statement as to whether your sample can be considered representative of a larger population, e) a description of how participants were recruited, and f) descriptions of where participants were recruited and where the research took place.

Response to the Reviewer: 

Thank you for the feedbacks. The recruitment process was done by screening for adolescents with obesity in seven public and three private high schools in the city of Yogyakarta, a city in the Southern part of Java, Indonesia. The participants were recruited at their schools and we screened 4268 students from 10 schools in Yogyakarta using three body mass index reference cut-off points: World Health Organization (WHO), Centre for Disease Control and Prevention (CDC), and International Obesity Task Force (IOTF). The screening was done from January to February 2016. We identified 298 (7%) adolescents classified as obese based on all those three obesity criteria. Of those, 229 (76.8%) obese adolescents agreed to participate in the next step of the study, i.e. assessment of their metabolic risk. The blood collection was taken at their schools, while the assessment of carotid intima media thickness was performed at the Dr Sardjito Hospital, Yogyakarta. These data collection was done from March to October 2016. 

Inclusion criteria were obese children aged 15 to less than 18 years. We excluded children with diabetes mellitus, renal disease, cardiovascular disease, any history of any systemic disease or history of current steroid use. This study included 156 (out of 229) obese adolescents who had data on CIMT. We considered that the sample of this study could be considered representative of a larger population. 

There were no differences in metabolic risk parameters between adolescents who had their CIMT measured versus those who had not (Appendix).

 "The study was partly funded by the Indonesian Pediatric Society and Frisian Flag Indonesia (355/Legal/FFI/XII/2014)".

i) Please provide an amended statement that declares *all* the funding or sources of support (whether external or internal to your organization) received during this study, as detailed online in our guide for authors at http://journals.plos.org/plosone/s/submit-now. Please also include the statement “There was no additional external funding received for this study.” in your updated Funding Statement. 

Response to the Reviewer: 

Thank you for the feedback. Regarding the funding of this study, we declare that the study was funded by the Indonesian Pediatric Society and Frisian Flag Indonesia (355/Legal/FFI/XII/2014). There was no additional external funding received for this study. 

ii) Please include your amended Funding Statement within your cover letter. We will change the online submission form on your behalf.

Response to the Reviewer: 

Thank you, the amended Funding Statement has been included in the cover letter. 

Response to the Reviewer: 

Done

Additional Editor Comments (if provided):

While the study had a lot of merit, there were a number of issues that need to be addressed. First their were significant grammatically errors throughout the text. Second, the data presented in the tables is unclear (what is being presented, skewed data is not indicated, and what was being compared). Third, why was Vitamin D not examined as a continuous variable? Also, from the raw data it appears that blood pressure was measured, why was this not included in the analysis? Lastly, as you did not find a association between vitD and CIMT, why was the analysis not extended to look at vitamin D to other metabolic risk factors measured.

While the study had a lot of merit, there were a number of issues that need to be addressed. First their were significant grammatically errors throughout the text. 

Response to the Editor:

The manuscript had been checked grammatically by a professional scientific editing service and this has been stated in the cover letter.

Second, the data presented in the tables is unclear (what is being presented, skewed data is not indicated, and what was being compared). 

Response to the Reviewer:

Thank you for the feedback. We have added explanation on the dependent and independent variables on the Tables and the skewed data have been clearly indicated. 

Table 1 describes the characteristics of obese adolescents from Yogyakarta stratified by sex. Normally distributed and skewed data are presented as mean (SD) and median (Q1;Q3), respectively. Normally distributed data are compared using independent sample t-test, while skewed data are compared using independent-samples Mann-Whitney U test. Categorical data are presented as n (%) and compared using chi-square tests unless otherwise indicated. 

Table 2 describes sex differences in the association of vitamin D and metabolic risk factors with carotid intima-media thickness in obese adolescents from Yogyakarta, Indonesia.

The dependent variable was CIMT as a marker of early vascular impairment. The independent variables were vitamin D and other metabolic risk factors including waist-to-hip ratio (marker of abdominal obesity), blood glucose, HbA1c (impaired glucose metabolism), fasting insulin, HOMA-IR (insulin resistance), LDL and HDL cholesterol, triglyceride (impaired lipid metabolism), blood pressure, smoking and family history of hypertension. 

Table 2 shows that overall, elevated HOMA-IR, total cholesterol, LDL-cholesterol and triglyceride levels were associated with greater odds of elevated CIMT. The associations persisted after adjustment for age, sex and second-hand smoke exposure. However, after the data were analysed within each sex, similar results were only observed in boys. None of the risk factors were associated with CIMT in girls.

Third, why was Vitamin D not examined as a continuous variable? 

Response to the Reviewer:

Thank you for the feedback. We actually are interested in looking at the clinical consequences of deficiency or insufficiency cut-offs of vitamin D status with the carotid intima media thickness as a measure of vascular thickness (a subclinical atherosclerosis in children). We used different cut off: Vitamin D level ≤30 ng/mL or ≤20 ng/mL. A categorisation of continuous data of vitamin D level was used in this study because it is easier to interpret and more feasible to present in clinical practice. It is helpful to label children and adolescents as having deficient, insufficient or normal vitamin D level. We hope that this paper is easier for clinicians to read and understand in order to make clinical decision making where decisions typical are categorical in nature. 

Also, from the raw data it appears that blood pressure was measured, why was this not included in the analysis? 

Response to the Editor:

We have included the analysis of blood pressure (as one of independent variables) and CIMT (as a dependent variable or outcome) at Table 2.

Lastly, as you did not find a association between vitD and CIMT, why was the analysis not extended to look at vitamin D to other metabolic risk factors measured.

Response to the Editor:

Thank you for the feedback. It seems like the introduction section of the paper was not clear enough to highlight why we are interested in looking at CIMT as predicted by vitamin D status independent of other predictors of CIMT. 

A paragraph has been added in the Introduction to further clarify the aim of the study. 

Carotid intima-media thickness (CIMT) is regarded as a reliable marker of subclinical atherosclerosis since increased CIMT can reflect an increase in arterial wall thickness including in children and adolescent (Atabek et al, 2014). Increased CIMT has been predicted in children with obesity, familial hypercholesterolemia, type 1 diabetes, hypertension (Giladini et al, 2011) and vitamin D deficiency (Atabek et al, 2014). In this study, we are particularly interested in looking at CIMT as predicted by vitamin D status independent of other predictors of CIMT.

The dependent variable was CIMT as a marker of early vascular impairment. The independent variables were vitamin D and other metabolic risk factors including waist-to-hip ratio (marker of abdominal obesity), blood glucose, HbA1c (impaired glucose metabolism), fasting insulin, HOMA-IR (insulin resistance), LDL and HDL cholesterol, triglyceride (impaired lipid metabolism), blood pressure, smoking and family history of hypertension. 

The association of vitamin D and other metabolic risk factors with CIMT was explored using logistic regression and presented as odds ratio (95% confidence interval). Each factor was tested in a separate regression model. Models that were significantly associated with elevated CIMT, were re-tested with adjustment for age, sex and secondary smoke exposures. All analyses were performed on a sample as a whole and also stratified by sex based on biological differences in body fat accumulation and potentially vitamin D levels between boys and girls. 

Reviewers' comments:

Reviewer's Responses to Questions

Comments to the Author

1. Is the manuscript technically sound, and do the data support the conclusions?

Reviewer #1: Yes

2. Has the statistical analysis been performed appropriately and rigorously? 

Reviewer #1: No

3. Have the authors made all data underlying the findings in their manuscript fully available?

Reviewer #1: Yes

4. Is the manuscript presented in an intelligible fashion and written in standard English?

Reviewer #1: Yes

5. Review Comments to the Author

Reviewer #1: Paper has lot of grammatical errors and flow is very limited

Need to assess high sensitivity CRP/?cardiac risk/LVMI and 24 hr ABPM or BP and would have correlated these parameters with those variables

Vitamin d level should be correlated with these parameters as we do not know what level is optimal one

urine albumin to creatinine ratio should reassesses

I will assess height and growth parameters 'also

need to develop or assess cardiovascular risk score like in adult to assess risk prediction

we should look into family history

It is association not causation

Need to assess high sensitivity CRP/?cardiac risk/LVMI and 24 hr ABPM or BP and would have correlated these parameters with those variables. Vitamin d level should be correlated with these parameters as we do not know what level is optimal one

Response to the Reviewer:

Thank you for the feedback. In this study, we particularly interested in looking at CIMT as predicted by vitamin D status independent of other predictors of CIMT. We are interested in looking at the clinical consequences of deficiency or insufficiency cut-offs of vitamin D status with the carotid intima media thickness as a measure of vascular thickness (a subclinical atherosclerosis in children).

We did not have data on hsCRP. But we did analyses on the association between CIMT (as an outcome) and vitamin D and other cardiovascular and metabolic risk factors including abdominal obesity (waist to hip ratio), blood glucose, HbA1c (glucose metabolism impairment), fasting insulin, HOMA-IR (insulin resistance), LDL and HDL cholesterol, triglyceride (dyslipidaemia), blood pressure, smoking, and family history of hypertension. We have added a paragraph in the Method section to clarify this.

We have added analysis on blood pressure and CIMT in the Table 2. But we did not include the analysis of LVMI in this study because we will do in another paper. 

urine albumin to creatinine ratio should reassesses

Response to the Reviewer:

Thank you for the feedback. We did not have data on albumin and creatinine ratio. 

I will assess height and growth parameters 'also

Response to the Reviewer:

Thank you for the feedback. We have analysis the height in term of waist-to-hip ratio between boys and girls in the Table 1 and the association between high waist-to-hip ratio with CIMT in the Table 2. But we did not analyse the association between growth with CIMT. We did not have the data of growth because of the nature of cross sectional study design. 

need to develop or assess cardiovascular risk score like in adult to assess risk prediction

we should look into family history 

Response to the Reviewer:

Thank you for the feedback. In this study we did not perform a cardiovascular risk score, we only assessed whether vitamin D and other metabolic risk factors predicted the increased CIMT without creating a score. But we have included a family history of hypertension as one of predictors of increased CIMT in this study. 

It is association not causation

Response to the Reviewer:

Thank you, noted.

---

## [Decision Letter · Decision Letter 1]

23 Nov 2020

PONE-D-20-19921R1

SEX DIFFERENCES IN THE ASSOCIATION OF VITAMIN D AND METABOLIC RISK FACTORS WITH CAROTID INTIMA-MEDIA THICKNESS IN OBESE ADOLESCENTS

PLOS ONE

Dear Dr. Murni,

Thank you for submitting your manuscript to PLOS ONE. After careful consideration, we feel that it has merit but does not fully meet PLOS ONE’s publication criteria as it currently stands. Therefore, we invite you to submit a revised version of the manuscript that addresses the points raised during the review process.

First I would like to apologize for the long delay, it was unfortunately as we had significant issues securing a reviewer for the manuscript.

I agree with the new reviewers comments that the manuscript is much improved, but analysis of VitD as a continuous variable should also be included. Regarding the table legends, it is still unclear what is being compared.  For table 1 the p value appears to be reporting significance comparing boys and girls? However, the p-values reported for table 2 are difficult to discern and are not explained clearly in the legend.

We look forward to receiving your revised manuscript.

Kind regards,

Jonathan M Peterson, Ph.D.

Academic Editor

PLOS ONE

Additional Editor Comments (if provided):

First I would like to apologize for the long delay, it was unfortunately as we had significant issues securing a reviewer for the manuscript.

I agree with the new reviewers comments that the manuscript is much improved, but analysis of VitD as a continuous variable should also be included. Regarding the table legends, it is still unclear what is being compared. For table 1 the p value appears to be reporting significance comparing boys and girls? However, the p-values reported for table 2 are difficult to discern and are not explained clearly in the legend.

Reviewers' comments:

Reviewer's Responses to Questions

**Comments to the Author**

1. If the authors have adequately addressed your comments raised in a previous round of review and you feel that this manuscript is now acceptable for publication, you may indicate that here to bypass the “Comments to the Author” section, enter your conflict of interest statement in the “Confidential to Editor” section, and submit your "Accept" recommendation.

Reviewer #2: (No Response)

2. Is the manuscript technically sound, and do the data support the conclusions?

Reviewer #2: Partly

3. Has the statistical analysis been performed appropriately and rigorously? 

Reviewer #2: No

4. Have the authors made all data underlying the findings in their manuscript fully available?

Reviewer #2: Yes

5. Is the manuscript presented in an intelligible fashion and written in standard English?

Reviewer #2: No

6. Review Comments to the Author

Reviewer #2: The carotid intima-media thickness (CIMT) is already well studied in children and adolescents, including assessing the relationship with low levels of vitamin D 25-OH.

The present study provides data from Indonesian adolescents, which is important to verify possible differences in this specific population.

The grammatical revision of the text really improved its quality. However, some aspects remain deficient:

1. Although the analysis of vitamin D as a categorical variable had been justified, it would be important, which is not so difficult, also analyze it as a continuous variable. Even if the result is negative, the analysis will be more complete. After all, the title, and the entire text focus on vitamin D. Otherwise, I suggest expanding the discussion to the other parameters evaluated, drawing attention to the positive findings. (Osika W, Dangardt F, Montgomery SM, Volkmann R, Gan LM, Friberg P. Sexual differences in the intimate peripheral artery, mean and thickness of the intimate media in children and adolescents. Atherosclerosis. 2009 March; 203 (1): 172-7 .)

2. Verification of normality uses the Kolmogorov-Smirnov test. In the text, “Smirnov” was replaced by a brand of vodka.

3. It is necessary to exclude the form of data presentation in the first columns, leaving only the name of the variable and the unit. The current format is redundant (the information is already described below the tables) and makes it difficult to understand.

7. PLOS authors have the option to publish the peer review history of their article (what does this mean?). If published, this will include your full peer review and any attached files.

Reviewer #2: No

---

## [Author Response · Author response to Decision Letter 1]

6 Jan 2021

Response to Reviewer

Thank you for the constructive questions and feedbacks. 

Additional Editor Comments (if provided):

First I would like to apologize for the long delay, it was unfortunately as we had significant issues securing a reviewer for the manuscript.

I agree with the new reviewers comments that the manuscript is much improved, but analysis of VitD as a continuous variable should also be included. Regarding the table legends, it is still unclear what is being compared. For table 1 the p value appears to be reporting significance comparing boys and girls? However, the p-values reported for table 2 are difficult to discern and are not explained clearly in the legend.

Thank you. The analysis of vitamin D levels as a continuous variable has been provided below:

Table 2. Correlations between of vitamin D levels and other metabolic risk factors to predict carotid intima-media thickness in obese adolescents (attached)

We found no correlation between vitamin D levels and CIMT, but we found a positive correlation between LDL and total cholesterols with CIMT. 

We also provide Table 3 showing the association of vitamin D levels and other metabolic risk factors to predict carotid intima-media thickness between boy and girl obese adolescents. Normally distributed continuous data are compared using independent sample t-test, while skewed data are compared using independent-samples Mann-Whitney U test. Categorical data are presented as n (%) and compared using chi-square tests unless otherwise indicated.

p value in the previous table 1 reported the significance comparing boys and girls. 

Odds ratio with its 95% CI and p value in the previous table 2 (now as Table 4) reported the significance of association of vitamin D levels and other metabolic risk factors to predict carotid intima-media thickness comparing boys and girls using logistic regression analysis.

Reviewer #2: The carotid intima-media thickness (CIMT) is already well studied in children and adolescents, including assessing the relationship with low levels of vitamin D 25-OH.

The present study provides data from Indonesian adolescents, which is important to verify possible differences in this specific population.

The grammatical revision of the text really improved its quality. However, some aspects remain deficient:

1. Although the analysis of vitamin D as a categorical variable had been justified, it would be important, which is not so difficult, also analyze it as a continuous variable. Even if the result is negative, the analysis will be more complete. After all, the title, and the entire text focus on vitamin D. Otherwise, I suggest expanding the discussion to the other parameters evaluated, drawing attention to the positive findings. (Osika W, Dangardt F, Montgomery SM, Volkmann R, Gan LM, Friberg P. Sexual differences in the intimate peripheral artery, mean and thickness of the intimate media in children and adolescents. Atherosclerosis. 2009 March; 203 (1): 172-7 .)

Response to the Reviewer:

Thank you. The analysis of vitamin D levels as a continuous variable has been provided in the new version of Table 2 below:

Table 2. Correlations between of vitamin D levels and other metabolic risk factors to predict carotid intima-media thickness in obese adolescents (attached)

We found no correlation between vitamin D and CIMT, but we found a positive correlation between LDL and total cholesterols with CIMT. 

We also provide Table 3 showing the association of vitamin D levels and other metabolic risk factors to predict carotid intima-media thickness between boy and girl obese adolescents. Normally distributed continuous data are compared using independent sample t-test, while skewed data are compared using independent-samples Mann-Whitney U test. Categorical data are presented as n (%) and compared using chi-square tests unless otherwise indicated.

The new table 4 showed association of vitamin D levels and other metabolic risk factors to predict carotid intima-media thickness comparing obese adolescent boys and girls using logistic regression analysis.

2. Verification of normality uses the Kolmogorov-Smirnov test. In the text, “Smirnov” was replaced by a brand of vodka.

Thank you, we have revised accordingly.

3. It is necessary to exclude the form of data presentation in the first columns, leaving only the name of the variable and the unit. The current format is redundant (the information is already described below the tables) and makes it difficult to understand.

Thank you, we have revised accordingly as suggested.

---

## [Decision Letter · Decision Letter 2]

15 Sep 2021

PONE-D-20-19921R2SEX DIFFERENCES IN THE ASSOCIATION OF VITAMIN D AND METABOLIC RISK FACTORS WITH CAROTID INTIMA-MEDIA THICKNESS IN OBESE ADOLESCENTSPLOS ONE

Dear Dr. Murni,

Thank you for submitting your manuscript to PLOS ONE. After careful consideration, we feel that it has merit but does not fully meet PLOS ONE’s publication criteria as it currently stands. Therefore, we invite you to submit a revised version of the manuscript that addresses the points raised during the review process.

We look forward to receiving your revised manuscript.

Kind regards,

Raffaella Buzzetti, M.D.

Academic Editor

PLOS ONE

Reviewers' comments:

Reviewer's Responses to Questions

**Comments to the Author**

1. If the authors have adequately addressed your comments raised in a previous round of review and you feel that this manuscript is now acceptable for publication, you may indicate that here to bypass the “Comments to the Author” section, enter your conflict of interest statement in the “Confidential to Editor” section, and submit your "Accept" recommendation.

Reviewer #3: All comments have been addressed

2. Is the manuscript technically sound, and do the data support the conclusions?

Reviewer #3: Yes

3. Has the statistical analysis been performed appropriately and rigorously? 

Reviewer #3: Yes

4. Have the authors made all data underlying the findings in their manuscript fully available?

Reviewer #3: Yes

5. Is the manuscript presented in an intelligible fashion and written in standard English?

Reviewer #3: Yes

6. Review Comments to the Author

Reviewer #3: Minor revisions are needed:

table 3: change the title of table 3. Authors did not check for "associations" but differences in patients with elevated CIMT and not elevated CIMT. Further, do not repeat statistical methods in the legend of the table, please revise also the sentence in tesxt (LDL cholesterol was associated with carotid intima media thickness among obese adolescent boys). Boys with elevated CIMT showed increased values of LDL cholesterol it is

7. PLOS authors have the option to publish the peer review history of their article (what does this mean?). If published, this will include your full peer review and any attached files.

Reviewer #3: No

---

## [Author Response · Author response to Decision Letter 2]

29 Sep 2021

Reviewer #3: Minor revisions are needed:

table 3: change the title of table 3. Authors did not check for "associations" but differences in patients with elevated CIMT and not elevated CIMT. Further, do not repeat statistical methods in the legend of the table, please revise also the sentence in tesxt (LDL cholesterol was associated with carotid intima media thickness among obese adolescent boys). Boys with elevated CIMT showed increased values of LDL cholesterol it is

Response to Reviewer:

Thank you so much for the advice, we have revised accordingly as suggested in the manuscript.

---

## [Editor Report · Decision Letter 3]

4 Oct 2021

SEX DIFFERENCES IN THE ASSOCIATION OF VITAMIN D AND METABOLIC RISK FACTORS WITH CAROTID INTIMA-MEDIA THICKNESS IN OBESE ADOLESCENTS

PONE-D-20-19921R3

Dear Dr. Murni,

We’re pleased to inform you that your manuscript has been judged scientifically suitable for publication and will be formally accepted for publication once it meets all outstanding technical requirements.

Kind regards,

Raffaella Buzzetti, M.D.

Academic Editor

PLOS ONE

---

## [Editor Report · Acceptance letter]

7 Oct 2021

PONE-D-20-19921R3 

Sex differences in the association of vitamin D and metabolic risk factors with carotid intima-media thickness in obese adolescents 

Dear Dr. Murni:

I'm pleased to inform you that your manuscript has been deemed suitable for publication in PLOS ONE. Congratulations! Your manuscript is now with our production department. 

Kind regards, 

on behalf of

Dr. Raffaella Buzzetti 

Academic Editor

PLOS ONE